# Tunable Synthesis of Predominant Semi-Ionic and Covalent Fluorine Bonding States on a Graphene Surface

**DOI:** 10.3390/nano11040942

**Published:** 2021-04-07

**Authors:** Jae-Won Lee, Seung-Pil Jeong, Nam-Ho You, Sook-Young Moon

**Affiliations:** 1Institute of Advanced Composite Materials, Korea Institute of Science and Technology (KIST), Chudong-ro 92, Bongdong-eup, Wanju-gun, Jeonbuk 55324, Korea; 092120@kist.re.kr (J.-W.L.); jsp0103@kist.re.kr (S.-P.J.); polymer@kist.re.kr (N.-H.Y.); 2Department of Advanced Materials Science and Engineering, Hanyang University, Ansan 15588, Korea

**Keywords:** fluorinated graphene, hydrothermal synthesis, semi-ionic bonding, tunable electronic properties

## Abstract

In this study, fluorinated graphene (FG) was synthesized via a hydrothermal reaction. Graphene oxides (GOs) with different oxygen bonding states and oxygen contents (GO(F), GO(P), and GO(HU)) were used as starting materials. GO(F) and GO(P) are commercial-type GOs from Grapheneall. GO(HU) was prepared using a modified Hummers method. The synthesized FGs from GO(F), GO(P), and GO(HU) are denoted as FG(F), FG(P), and FG(HU), respectively. The F atoms were bound to the graphene surface with predominantly semi-ionic or covalent bonding depending on the GO oxygen state. FG(F) and FG(HU) exhibited less extensive fluorination than FG(P) despite the same or higher oxygen contents compared with that in FG(P). This difference was attributed to the difference in the C=O content of GOs because the C=O bonds in GO primarily produce covalent C–F bonds. Thus, FG(F) and FG(HU) mainly exhibited semi-ionic C–F bonds. The doped F atoms were used to tune the electronic properties and surface chemistry of graphene. The fluorination reaction also improved the extent of reduction of GO.

## 1. Introduction

Graphene is a two-dimensional (2D) material that has attracted considerable interest because of its extraordinary properties, such as high electrical conductivity, thermal conductivity, and mechanical strength [1,2,3,4]. Graphene shows different properties with its layer number. The single-layer graphene with no defects has zero bandgap material. However, doping with heteroatoms, such as nitrogen [5,6], boron [7,8,9], and sulfur [10,11] on a graphene surface can tune the structure and electrical, optical, and thermal properties. Among these graphene derivatives, fluorinated graphene (FG) exhibits unique chemical, physical, mechanical, and electrical properties compared to graphene [12,13,14,15,16,17,18]. In particular, the bandgap of FG can be continuously controlled from 0 to 3.4 eV with F doping amounts [19]. In contrast, the F atoms introduced in the graphene surface changed the hybridization of C–C bonds from sp^2^ to sp^3^ [20]. The C–F bond showed different binding characteristics with semi-ionic and covalent bonding. The semi-ionic C–F bonding selectively reduced GO and recovered the electrical conductivity. However, covalent C–F bonding is more stable and has a higher bonding strength (E_disso_ = 460 kJ/mol), which shows that FG is an insulator. Therefore, the binding state of C–F bonding is closely related to the electrical and thermal properties. FG can be synthesized via several methods, including XeF_2_ gas treatment [20,21], exfoliation of bulk fluorinated graphite [17,22,23], and the hydrothermal process with HF [24]. The hydrothermal process is a simple and effective method for the scalable production of FG because of its controllable chemical modification. The hydrothermal method used GO dispersant. GO is a good starting material for hydrothermal reactions because of many oxygen-containing groups, such as epoxide, carboxylic, and ketones. The oxygen-containing groups can be reduced and substituted for a C–F bond. However, the uniform FG from a hydrothermal reaction has not been reported yet. Because the chemical state of the GO surface is not uniform, it is difficult to control the C/F ratio or the bonding state of F. Furthermore, the major factors affecting the fluorination have not yet been clarified. Therefore, determining the factors affecting the fluorination reaction is important for the development of future scalable manufacturing methods and various applications.

In this study, FG was synthesized via a hydrothermal reaction. The prepared FGs exhibited different C/F atomic ratios and controllable C–F bonding states with the predominant semi-ionic or covalent bonding. Most factors that influence the adjustment of the C–F bond state depend on GOs’ initial chemical state. This difference was attributed to the difference in the C=O content of GOs because the C=O bonds in GO primarily produce covalent C–F bonds. The relationship between the chemical composition of GO and C–F bonding state is discussed in detail.

## 2. Materials and Methods

### 2.1. Materials and Synthesis of FG

Three types of GO with different oxygen groups were used in this study. Type I (GO(F)) and type II ((GO(P)) with different oxygen states were purchased from Grapheneall. Type III ((GO(HU)) was prepared using a modified Hummers method [25]. Briefly, 5 g of graphite (Daejung, industrial grade) and 2.5 g of NaNO_3_ were stirred with 115 mL H_2_SO_4_ in an ice bath. Then, 15 g of KMnO_4_ was slowly added to the solution. Over the next 4 h, the reaction solution was removed from the ice bath and continuously stirred until it reached room temperature. The solution was heated at 35 °C for 30 min and then mixed with 250 mL of deionized (DI) water, followed by heating to 70 °C for 15 min. After this, the solution was poured into 1 L of DI water, and 15 mL of 3% H_2_O_2_ solution was added to remove unreacted KMnO_4_ and MnO_2_. The synthesized GO was purified by centrifugation at 10,000 rpm for 30 min and dispersed in DI water repeatedly. The obtained GO slurry was then dried in a vacuum overnight.

FG was synthesized via a simple hydrothermal reaction [24]. In a typical procedure, 11 mg of GO was dispersed in 12 mL DI water by ultrasonication (pulse 10 s on and 5 s off, amplitude 20%, 20 kHz, 500 W, VC500, Sonics & Materials, Newtown, CT, USA) for 30 min under 10 °C. HF (40 wt %, 3.2 mL) was added and mixed. The solution was poured into a 50 mL Teflon-lined autoclave and stirred at 230 °C for 12 h. After the reaction, a white product was obtained in the solution, which was FG. The as-prepared FG was filtered with 0.45 µm micropore and washed with DI water several times and then freeze-dried at −100 °C for 48 h. The synthesized FG was denoted as FG(F), FG(P), and FG(HU) according to GOs.

### 2.2. Materials Characterization

The structure of synthesized FG was characterized using Raman spectroscopy (Renishaw, InVia; 514 nm excitation laser, Wotton-under-Edge, UK), transmission electron microscopy (TEM, 30 kV, Titan CUBED 60-300, FEI, Hillsboro, OR, USA), atomic force microscopy (AFM, Park NX10, Park systems, Suwon, Korea) with a non-contact mode, and X-ray diffractometer (XRD, Cu Kα, λ = 0.154 nm, 45 kV, 200 mA, SmartLab, Rigaku, Akishima-shi, Tokyo, Japan). The compositions of FGs were determined using energy-dispersive X-ray spectroscopy (EDS, QUANTAX, Bruker, Billerica, MA, USA), Fourier-transform infrared spectrometry (FTIR, FTIR-6600, JASCO, Hachioji, Tokyo, Japan), and X-ray photoelectron spectroscopy (XPS, monochromatic Al Kα, Thermo Fisher Scientific, Waltham, MA, USA). The thermal behavior was analyzed through thermogravimetry–differential scanning calorimetry (TG–DSC, Labsys Evo, Setaram, Geneva, Switzerland) with a heating rate of 10 °C/min to 800 °C in air.

## 3. Results

FG was synthesized with a tunable chemical bonding state with semi-ionic and covalent bonding. All the synthesis conditions of FGs were the same except for the GOs having different O contents and binding states.

The microstructure of the synthesized FG was characterized using high-resolution transmission electron microscopy (HRTEM) (Figure 1). A typical 2D sheet-like structure was observed. FG exhibited an irregular atomic arrangement because of the F dopant. The edge structure showed few-layer stacking, and the thickness of all of the FG sheets was ~4 nm, as observed through the AFM images (Appendix A
Appendix A). The AFM images also indicated that FGs should be 3–4 layers thick. FG(P) exhibited a highly wrinkled structure compared with those of the other samples, indicating that FG(P) had a higher F content than those of other FGs. The morphologies of all the GOs exhibit similar thickness and size (Appendix A). Thus, it seems that the different morphological appearance of FG(P) is a result of the C–F bond structure. This relationship between the chemical state of GO and the bonding structure of C–F is discussed later. The distribution of doped F atoms on the graphene surface was investigated using EDS. FG(P) exhibited the maximum F distribution, which is considered to affect structural morphology. The hybridization from sp^2^ to sp^3^ results in stress in the graphene layer, which causes disorder and buckling of the structure.

The fluorination and reduction processes were further confirmed using Raman spectroscopy (Figure 2), which is useful for characterizing both the atomic structure and electronic properties of GO and FG [26]. The G band arises from the disorder scattering of sp^2^ carbon atoms, whereas the D band is associated with atomic-scale defects and lattice disorder. All of the GOs showed similar peak positions of the D band (1355 cm^−1^) and G band (1596 cm^−1^). The I_G_/I_D_ ratios were 1.134 (GO(F)), 1.112 (GO (P)), and 0.979 (GO(HU)) (Figure 2a). The structure of GOs was also characterized using XRD (Appendix A). The interlayer distance was calculated using Bragg’s equation. All the GOs showed typical GO peaks. The sharp peaks of GO(F), GO(P), and GO(HU) indexed to (001) appeared at 10.05°, 10.25°, and 10.8°, respectively. The second peak at 42.6 of GOs is attributed to (100). The calculated interlayer distances were 0.878, 0.862, and 0.819 nm for GO(F), GO(P), and GO(HU), respectively. GO(HU) showed a lower interlayer distance than GO(F) and GO(P). It is suggested that GO(HU) has a low number of oxygen groups on its surface.

The shift in the peak position on the D band and G band determined the structure deformation. The G band of FG(P) showed a significant blue shift to 1612 cm^−1^ caused by the structural imperfections arising from the attachment of F atoms on the graphene layer compared with 1598 cm^−1^ of FG(F) and 1582 cm^−1^ of FG(HU) (Figure 2c). Only FG(HU) showed red-shift because of reduction in and the self-healing effect of thermal treatment. FG(P) was heavily fluorinated; thus, the graphene structure had a sufficient concentration of defects, as confirmed through HRTEM. As a result, the I_G_/I_D_ ratio of GO(P) decreased to 0.816. Unlike GO(P), the I_G_/I_D_ ratios of FG(F) and FG(HU) increased to 1.163 and 1.047, respectively, because of the reduction in GO with high temperature.

The electronic configurations of the FGs were analyzed using UV-vis spectroscopy (Appendix A). The absorbance of the FGs changed significantly compared with that of GO. The spectrum of pristine GO showed a wavelength of maximum absorption at 230 nm, which is associated with the characteristic π–π * electron transition [27]. After fluorination and reduction, the peak red-shifted to ~260 nm, which indicates an increase in the π-electron concentration and the structural order due to the possibility of restoration and atomic rearrangement of the sp^2^ carbon, indicating GO reduction [28].

Figure 3 shows the FTIR spectra of the GOs and FGs. For the spectra recorded for all GO samples (Figure 3a), distinct and strong peaks were observed at approximately 3200–3500 cm^−1^ as a result of the −OH stretching vibrations of the hydroxyl groups. The peaks located at approximately 1726 cm^−1^ were attributed to the C=O stretching vibrations of the carbonyl and carboxylic groups [26,29,30,31]. GO(P) showed a sharp peak of C=O compared to the other two GOs, indicating a high content of C=O in GO(P) compared with the other GOs. The C=O contents increased in the order GO(P) > GO(F) > GO(HU). GO(HU) exhibited the lowest C=O. In addition, the peaks located at approximately 1617 and 1398 cm^−1^ were attributed to the C=C stretching mode of the non-oxidized graphitic domains and the deformation vibration of tertiary–OH groups, respectively. Moreover, the peaks at 1227 and at 1049 cm^−1^ were attributed to the stretching vibrations of C–O–C and C–O, respectively [32]. GO(P) had a higher number of oxygen-containing groups than GO(F) and GO(HU). These differences in the number of oxygen groups resulted in the fluorination of GO. The spectrum of FG showed decreased hydroxyl and oxygen-containing functional groups (Figure 3b). The C–F peaks appeared between 1000 and 1150 cm^−1^. The vibration mode at 1080 cm^−1^ corresponded to C–F semi-ionic bonding [33,34,35]. FG(F) and FG(HU) showed higher peak intensities at 1080 cm^−1^, implying that there are predominant semi-ionic bonds in FG(F) and FG(HU) compared with FG(P). Moreover, FG(P) showed the increased intensity of the peak at 1222 cm^−1^, which can be attributed to the CF_2_ stretching vibration with covalent bonding [36]. The F atom was bonded with the completely sp^3^-hybridized carbon atom in FG(P). A band at 1650 cm^-1^ attributed to C=C due to the planar graphene layer structure obviously decreased in FG(P). This suggested that all the graphene layers are arranged in an sp^3^-type puckered structure. This result matches the TEM results.

To further investigate the surface properties, FG was analyzed using X-ray photoelectron spectroscopy. Figure 4 shows the F 1s peaks in the XPS spectra of the FGs to determine the C–F bonding state. The peak was analyzed using a peak analyzer via origin software. Curve fitting of the spectra was performed using a Gaussian peak fitting after conducting a Shirley background correction. As shown in Figure 4, the F 1s peak in the spectrum of the FG sample was strong, indicating that F was bonded well onto graphene via the hydrothermal reaction, and the O content in all the FGs decreased as a result of chemical reduction. The spectra of FG(F) and FG(P) clearly showed three peaks. The fitted peaks corresponding to the C–F semi-ionic bonds, C–F, and C–F_2,_ appeared at 685.8, 687, and 689 eV, respectively [37]. However, FG(HU) showed only two peaks, semi-ionic and C–F, and there was no C–F_2_, which indicated that there was less fluorination than in the other two samples. The spectra of FG(F) and FG(HU) showed clear peaks that can be attributed to semi-ionic C–F, and the spectrum of FG(P) exhibited prominent peaks associated with the covalent bonds. As mentioned above, GO(HU) showed low C=O and C–O–C bonding on the graphene plane compared with GO(F) and GO(P), as observed in the FTIR spectra. Therefore, the covalent bonding of C–F_2_ was not found.

The C 1s spectra were also obtained to confirm the reduction and fluorination of FG (Figure 5). There were mainly six fitted peaks: 284.3 (C–C), 285.2 (C–O), 286.5 (C=O), 287.6 (C–F semi-ionic bonds), 289.2 (C–F), and 290.4 eV (C–F_2_) [38,39,40]. The three F-binding bonds in the FG samples were semi-ionic C–F (287.6 eV), C–F (289.2 eV), and C–F_2_ (290.4 eV), with small fractions of C–F_3_ (293.7 eV). The C–F_2_ and C–F_3_ peaks were most predominant in the spectrum of FG(P). The degree of fluorination of FG can be estimated based on the F-to-C content ratio (F/C), as determined from the C 1s spectra by summing the F content in each F-containing group [41]. FG(P) exhibited the highest fluorination among the investigated FGs, with an F/C ratio of 1.47. The F/C ratio decreased in the order FG(P) (1.47), FG(F) (0.49), and FG(HU) (0.3). The synthesized FG(P), FG(F), and FG(HU) had compositions of C_0.41_F_0.59_, C_0.67_F_0.33_, and C_0.79_F_0.21_, respectively. Thus, the fluorination degree of FG can be adjusted according to the initial graphene chemical structure.

The reduction and fluorination rates were closely related to the oxygen bonding state of GO (Appendix A). The C 1s spectra of GO clearly showed five peaks at ~284 eV (C=C), 284.8 eV (C–C), 286.5 eV (C–O), 287.1 eV (C=O), and 288.7 eV (O–C=O). The O content as various bonding states decreased clearly decreased, particularly for C=O. This indicated that O was replaced by F and doped on the graphene surface such that the reduction reaction proceeded simultaneously [21,37]. FG(F) and FG(P) exhibited higher C=O contents than FG(HU). As a result, the fluorination rate of FG(HU) was confirmed to be lower than those of others. The substitution reaction between the F and O functional groups (C–O–C/C=O) of GO with sp^3^-hybridized C atoms resulted in covalent C–F bonds. In contrast, the carboxyl groups (-COOH) tended to detach easily from the GO surface during the hydrothermal process because of instability, subsequently leaving active sites with sp^2^-hybridized C atoms; these active sites formed semi-ionic C–F bonds. FG(F) had a higher carboxyl group and C–OH than FG(P), which led to the formation of semi-ionic C–F, which reacted quickly even with similar C=O. The amount of C=O affected the formation of the covalent bonds; C–OH and carboxyl groups affected the formation of the semi-ionic bonds. This played an important role in determining the mechanism of the fluoridation reaction. Thus, we synthesized FG with a tunable C/F atomic ratio and a tunable chemical bonding state by varying the initial chemical state of graphene. It was found that the bonding states of the oxygen-containing groups influenced fluorination.

The greater covalence of the C–F bonds can enhance the chemical and thermal stability of fluorinated carbon materials. The thermal stability of FG at temperatures as high as 800 °C was determined by thermogravimetric analysis (Figure 6). The significant weight loss under 200 °C of GOs (Figure 6a) can be attributed to the elimination of the water molecules and unstable oxygen groups. Decomposition at 200 °C is caused by the elimination of more stable oxygen functional groups and the decomposition of carbon. The decomposition shown by GO(P) and GO(F) occurred more rapidly than that of GO(HU). This finding suggested that GO(F) and GO(P) have many more unstable oxygen groups than GO(HU). After fluorination, the thermal stability of FG was improved substantially (Figure 6b). FG began to decompose in the temperature range of 300–400 °C. In decomposition at higher temperatures (400–600 °C), FGs are converted into low-molecular-weight volatile compounds (C_x_F_y_). FG shows good thermal stability as a result of strong C–F bonding energy. FG(P) shows higher weight loss compared to others. It suggests that the structure is unstable due to high fluorination and is easily changed to a volatile compound (CxFy) [42]. On the other hand, FG (F) and FG (HU) have a relatively stable structure compared to FG (P) and are considered to maintain the advantages of fluorination while maintaining thermal stability. The fluorination degree was determined by the oxygenation state, which strongly influenced the chemical composition and structure of the final FG. Consequently, it was confirmed that the chemical properties of graphene affect the fluorination and, conclusively, the thermal properties.

## 4. Conclusions

In summary, FG was synthesized with tunable C/F contents and a tunable chemical bonding state with semi-ionic and covalent bonding via a hydrothermal method. The difference in the O content and chemical states on the GO surface resulted in different extents of fluorination and reduction. The fluorination reaction improved the degree of reduction of GO. A high concentration of C=O in GO promoted the formation of covalent bonds of C–F, C–F_2_, and C–F_3_ in FG. Oxygen in other states, such as –COOH, resulted in semi-ionic bonding. The reaction mechanism of fluorination was determined by tracking the oxygen bonding state in the initial GO. The doped F atoms tuned the electronic and thermal properties and surface chemistry of graphene. Highly FG exhibited high thermal stability, thereby making it a promising candidate for use in various applications.

## Figures and Tables

**Figure 1 nanomaterials-11-00942-f001:**
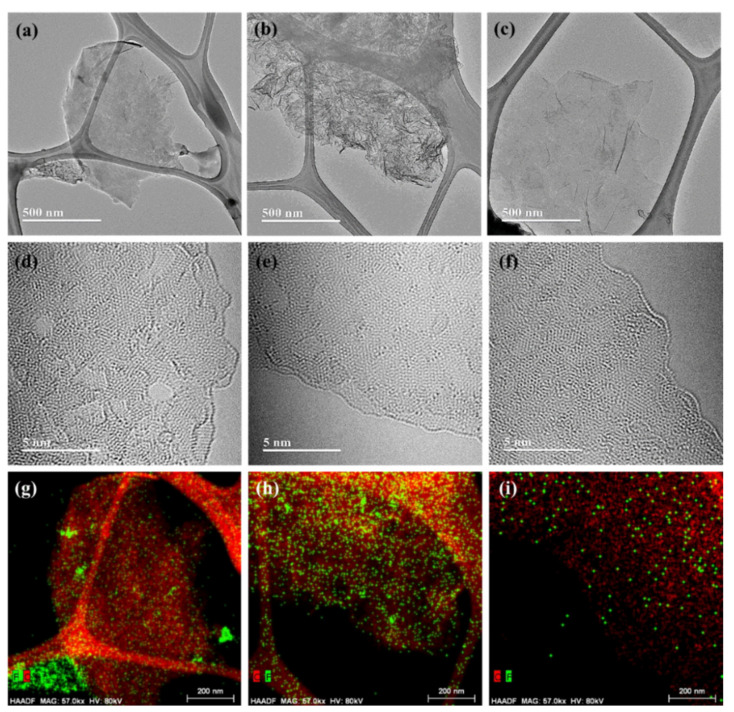
Microstructure of synthesized fluorinated graphene (FG) and distribution of doped F on FG. High-resolution transmission electron microscopy (HRTEM) images of FG: (**a**,**d**) FG(F), (**b**,**e**) FG(P) and (**c**,**f**) FG(HU). Energy-dispersive X-ray spectroscopy (EDS) images of C (red dot) and F (green dot): (**g**) FG(F), (**h**) FG(P) and (**i**) FG(HU).

**Figure 2 nanomaterials-11-00942-f002:**
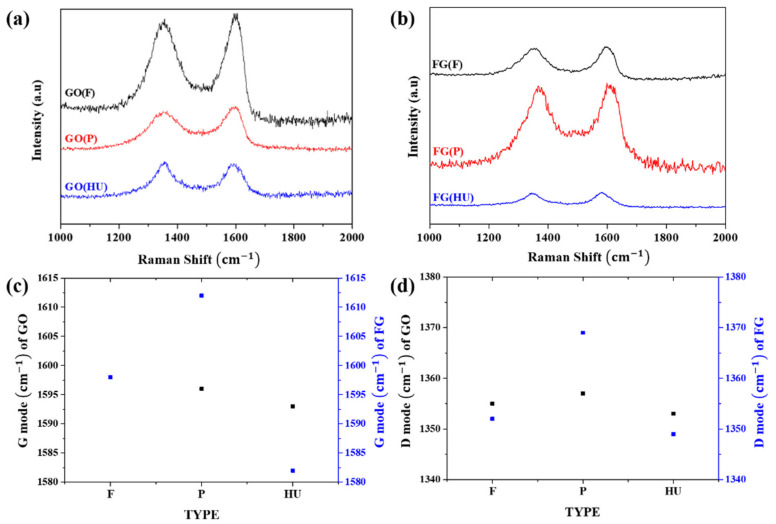
Raman spectra of graphene oxide (GO) and synthesized FG: (**a**) GO, (**b**) FGO, (**c**) D band, and (**d**) G band.

**Figure 3 nanomaterials-11-00942-f003:**
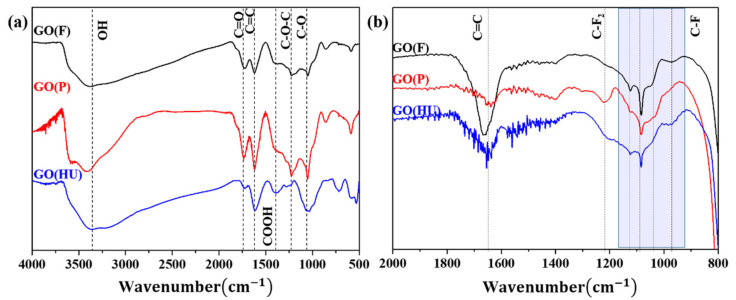
FTIR spectra of (**a**) GO and (**b**) FG.

**Figure 4 nanomaterials-11-00942-f004:**
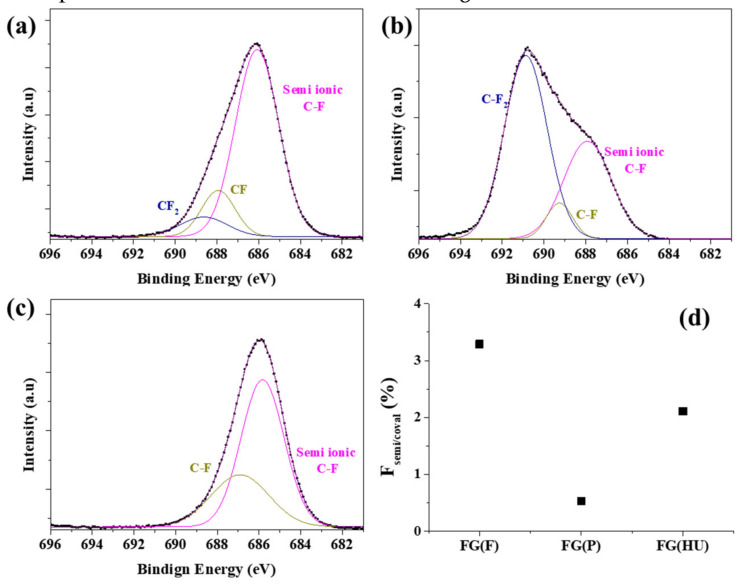
XPS spectra of the F 1s peaks of fluorinated graphene: (**a**) FG(F), (**b**) FG(P), (**c**) FG(HU), and (**d**) the ratio of semi-ionic/covalent C–F.

**Figure 5 nanomaterials-11-00942-f005:**
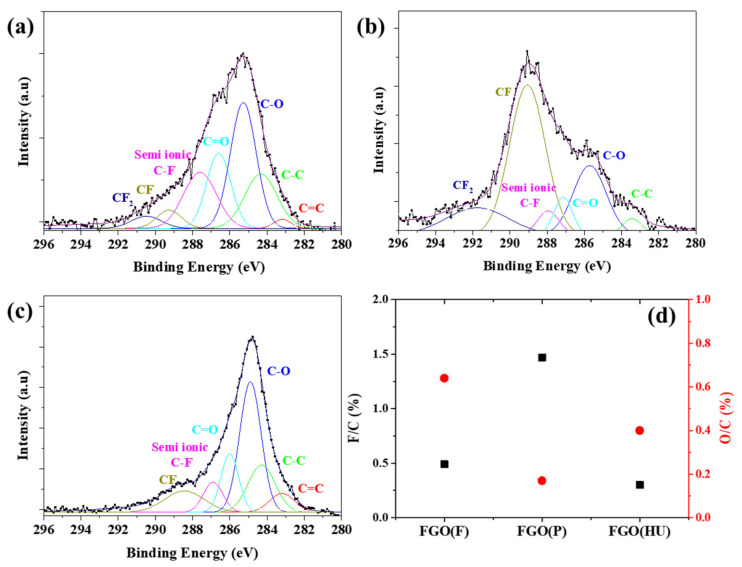
XPS spectra of C 1s of fluorinated graphene: (**a**) FG(F), (**b**) FG(P), (**c**) FG(HU), and (**d**) the ratio of F/C and O/C.

**Figure 6 nanomaterials-11-00942-f006:**
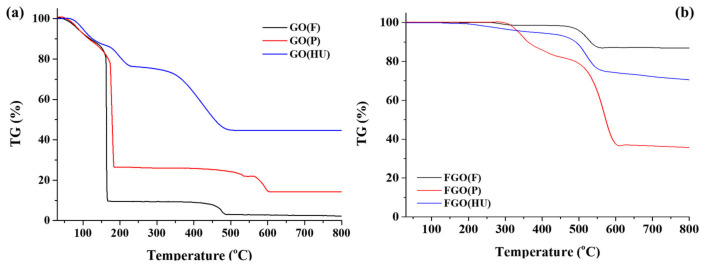
Thermal behavior of GO (**a**,**b**) FG obtained using thermogravimetric analysis.

## Data Availability

No new data were created or analyzed in this study. Data sharing is not applicable to this article.

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
