# Peer review of "Tunable Synthesis of Predominant Semi-Ionic and Covalent Fluorine Bonding States on a Graphene Surface"

_nanomaterials, 2021, doi:10.3390/nano11040942_

Round 1
Reviewer 1 Report
This manuscript submitted to publication to Nanomaterials by Lee et al. reports on the synthesis and characterization of fluorinated graphenes from three different graphite oxide sources. The manuscript has two major claims:
- „An effective synthesis method…is demonstrated for the first time in the present study”.
- „The chemical composition and morphology of the product and the fluorination mechanism of the reaction are investigated and discussed.”
Out of these two major points, I think that the first one is completely refutable. The authors show themselves that „FG was synthesized via a simple hydrothermal reaction” as found ref. 12. In this way, the synthesis is certainly not demonstrated first time. Regarding the second claim, indeed, the chemical states of the fluorine and oxygen-bonded carbon atoms are characterized by detailed XPS measurements. Here, the spectral features are very likely well measured (though we cannot check about experimental details). At the same time, the interpretation of spectra may not be that obvious (explained later). Therefore, I think that the quality and especially, the novelty of the manuscript does not meet the standards of the Q1 journal „Nanomaterials”, and should be rejected in this form. I suggest the authors either to (1) restructure and carefully rewrite the manuscript focusing on the relation of the GO chemistry vs. Fluoride bonding in more details (if the Editors allow resubmission despite the serious flaws) or (2) resubmit to a more specified journal focusing on spectroscopic features, and less focusing on nanomaterial’s science.
Regarding the first option: the main problem with the characterization is that the authors strongly rely their discussion on the C=O content of GO (“The high concentration of C=O in GO promotes the formation of covalent bonds of C–F, 194 C–F2, and C–F3 in FG”. However, the C=O assignment in XPS (and therefore the C=O content) is not well established. It is relatively well known that while deconvoluted peak assignments are possible for carbon C1s envelopes, their interpretation is often ambiguous. The authors are certainly not responsible for this but small changes in the peak positions can cause large changes in the relative areas of the bond types (C=C, C-OH, C=O, COOH).
Therefore, the authors are suggested to have additional information on the relative C=O content of their materials by IR spectroscopy, which is the best technique for the analysis of carbonyl vibrations in graphite oxide and thermally decomposed derivatives (see e.g. or DOI: 10.1016/j.carbon.2005.07.013 or https://doi.org/10.1021/jp2052618 and references therein)
**********************
Other major problems:
SCIENCE:
- the authors basically do not use subscripts and superscripts, which make the look of manuscript very “sloppy”: for example, “sp2 to sp3” or XeF2, KMnO4 etc…
- the hydrothermal procedure is not written correctly, because it cannot be reproduced without mentioning the amount of solvent: “In a typical procedure, 11 61 mg of GO was dispersed in DI water by ultrasonication.” – How much water was used? It affects composition and thus possibly the formation rate of the products.
- few details are provided for “materials characterization”: For example, how was sample done for AFM, etc. Was it contact mode, or noncontact mode etc.?
- XRD is a crucial technique for checking the transformation of graphite compounds - why was not it used to evidence the reaction?
- What does it mean: „The thermodynamic reaction was analyzed”? Thermodynamics characterizes the energy changes of all reactions.
- „The hybridization from sp2 to sp3 leads to longer C–C bonding length, which lead to disorder and buckling of the structure.” Can the authors justify or clarify this sentence? I think that disorder and buckling happens not only because of the (slight) change of the bond lengths, but because of the difference in the geometrical arrangement of atoms around sp2 (planar trigonal) and a sp3 (tetrahedral) carbon atoms.
- „The spectra of FG(F) and FG(P) clearly appear three peaks.” I do not clearly see three peaks on the spectra. Of course, authors present three peak components, which are clearly resolved, but the experimental spectrum does not show three peaks.
- Figure 3d: F/C and O/C ratios are shown in percentages (%)? Eg. 0.4% O/C? Is that correct?
- What is the meaning of “The greater covalence of C–F bonds”?
STYLE: the manuscript is not written carefully at all. I suggest that all coauthors re-read the manuscript before submission. There is a great number of basic mistakes in English such as:
- “there are also tried to”
- “until it researched room temperature” makes no sense
- “FG(P,F and HU) abbreviations are explained in the text, but not in the abstract. The latter is self-standing, so it must be explained there also (for all abbreviations).
- “F was well bonded onto the graphene”: what does “well” mean here? “Strongly bonded” may be better.
Author Response
=> The hydrothermal process is a simple and an effective method for scalable production FG because of its controllable chemical modification. And the GO is good starting materials for hydrothermal reaction because of many oxygen containing groups, such as epoxide, carboxylic and ketones. But the uniform FG from hydrothermal reaction has not reported yet. Because the chemical state of the GO surface is not uniform, it is very difficult to control the C/F ratio or the bonding state of F. And the major factors affecting the fluorination have not yet been clarified. In this study, the relationship between the chemical composition of GO and C-F bonding state are discussed in details. Therefore, this manuscript is very meaningful as it reveals the factors that have a decisive influence on the fluoridation reaction. This is very important for the development of manufacturing methods and various applications for future industrialization..
**********************
SCIENCE:
the authors basically do not use subscripts and superscripts, which make the look of manuscript very “sloppy”: for example, “sp2 to sp3” or XeF2, KMnO4 etc…
the hydrothermal procedure is not written correctly, because it cannot be reproduced without mentioning the amount of solvent: “In a typical procedure, 11 61 mg of GO was dispersed in DI water by ultrasonication.” – How much water was used? It affects composition and thus possibly the formation rate of the products.
=> We apologize for this problem. We revised the experimental section and the details can confirm in the manuscript.
few details are provided for “materials characterization”: For example, how was sample done for AFM, etc. Was it contact mode, or noncontact mode etc.?
=> We revised the characterization section and the details can confirm in the manuscript.
XRD is a crucial technique for checking the transformation of graphite compounds - why was not it used to evidence the reaction?
=> We appreciate reviewer’s comment. We investigated the structure by XRD and the details can confirm in the manuscript.
What does it mean: „The thermodynamic reaction was analyzed”? Thermodynamics characterizes the energy changes of all reactions.
=> We apologize for confusion. The thermal behavior analyzed TG-DSC. We have corrected the expression and details can be found in the manuscript.
„The hybridization from sp2 to sp3 leads to longer C–C bonding length, which lead to disorder and buckling of the structure.” Can the authors justify or clarify this sentence? I think that disorder and buckling happens not only because of the (slight) change of the bond lengths, but because of the difference in the geometrical arrangement of atoms around sp2 (planar trigonal) and a sp3 (tetrahedral) carbon atoms.
=> We apologize for confusion. We mean that the hybridization from C=C (sp2) to C-C (sp3) leads to change bonding length, which lead to disorder and buckling of the structure because of stressed in the graphene layer. When other elements are bonded to the graphene layer, the atoms constituting the existing structure are rearranged and stress is applied to the bonds of each element. Therefore, the length of each bond changes, which can be confirmed in the Raman results.
„The spectra of FG(F) and FG(P) clearly appear three peaks.” I do not clearly see three peaks on the spectra. Of course, authors present three peak components, which are clearly resolved, but the experimental spectrum does not show three peaks.
=> We carefully performed the peak analysis through Origin program, which was confirmed through several verifications. As you know, the various combinations hidden in the broad XPS spectra are separated and analyzed by an analysis program. We analyzed by a professional analysis program, and I think the results are well included in the paper.
Figure 3d: F/C and O/C ratios are shown in percentages (%)? Eg. 0.4% O/C? Is that correct?
=>The ratio is percentage.
What is the meaning of “The greater covalence of C–F bonds”?
=> We apologize for confusion. It means that the covalent bonding is predominant. But we have corrected the expression and details can be found in the manuscript.
STYLE: the manuscript is not written carefully at all. I suggest that all coauthors re-read the manuscript before submission. There is a great number of basic mistakes in English such as:
“there are also tried to”
“until it researched room temperature” makes no sense
“FG(P,F and HU) abbreviations are explained in the text, but not in the abstract. The latter is self-standing, so it must be explained there also (for all abbreviations).
“F was well bonded onto the graphene”: what does “well” mean here? “Strongly bonded” may be better.
=>We apologize for this problem. The manuscript was edited by professional English editing service with native speaker. Details can be found in manuscript.

Reviewer 2 Report
Paper entitled “Tunable Synthesis of Predominant Semi-Ionic and Covalent Fluorine Bonding States on a Graphene Surface” reports the synthesis of fluorinated graphene (FG) employing a hydrothermal route and show that the added F atoms tune the electronic properties of the resulting FG. The abstract could be improved, using “graphene” alone is confusing, authors should explain right away that graphene refers to FG, a type of reduced graphene (thermally reduced from GO). Otherwise, at the first glance, the reader gets the impression that authors attached F atoms to “pure” CVD-like graphene. The characterization methods are appropriate and efficient. Experiments are concise and systematically planned and performed, results are interesting, useful for research community dealing with graphene-like materials. Conclusions are supported by the experiments and references are rather fair. I would recommend the paper to be published in Nanomaterials, but with some improvements that have to be made, as indicated below.
- In the abstract, line 10, graphene should be used as “reduced graphene” (or “the GO surface”?), in order to avoid further misunderstandings;
- FG(F) and FG(HU) needs to be defined or explained in the abstract, right now it is not possible to understand the abstract right away, without reading the main text;
- Line 21, “it” is not needed;
- I have red with pleasure the introduction part, which is nice, concise, not lengthy and still comprehensible. Nonetheless, some statements could be strengthened by additional references. I believe it could be improved to offer the readers a more perspective view on graphene varieties. Therefore, a reference or two after “properties” in line 22 are needed (maybe Science 2008, 321, 385-388; and/or Solid State Commun. 2008, 146, 351-355 or similar) so that all readers can find those properties and learn about them (otherwise how can the reader compare these properties with respect to properties of F-reduced-graphene in lines 24-26); the paper is intended for all readers not only experienced researchers who already have that kind of information. Furthermore, the expression “Among the numerous graphene derivatives” has to be more explanatory and followed by references, otherwise it is just a very general and meaningless statement with no support from the literature. Maybe: “Among the numerous graphene derivatives, that include GO [Ref: ACS Appl. Mater. Interfaces 2017, 9, 43393-43414 or similar] and RGO [ Mater. Chem. C, 2020, 8, 1198 or similar], fluorinated graphene (FG) is of interest because …..”. Authors should avoid using twice “attracting”, as it is “diluting” the quality of the text. Important: before starting the second sentence “The F atoms ….” in line 23, the authors should add a short, concise sentence explaining how FG, a reduced type of graphene [use Ref: 12 from your text or similar], is different from a “regularly” non-fluorinated/reduced graphene oxide [use Ref: Synthetic Metals, 2020, 269, 116576 or similar].
- “Along with the synthesis, there are also tried to 33 control the degree of fluoridation” needs reformulation.
- H2SO4 should be H2SO4 in line 53; see also for H2O2, KMnO4and MNO2, etc.’
- “was removed” in line 54;
- Reformulate: “After reaction, the white product synthesized in solution, which is FG.”;
- Reformulate: “The C 1s spectra also observed to confirm reduction and fluorination of FG (Fig. 3).”;
- It looks bad to use in same text Figure, Fig. and FiG. Please fix this through the whole text;
- There is no text describing/referring to results in figure 2d;
- Sentence “FG(F) and FG(P) exhibit higher C=O contents than FG(HU).” in line 136 seems to contradict with Figure 3. Double check etxt, FG(P) seems to have a smaller C=O peak! Double check the caption of Figure 3, (c) and (d) are missing, (d) is not described at all. Same for Figure 2.
- “lower than” in line 137;
- Again, there is no text referring/describing Figure 3d;
- Reformulate: “The thermal stability of FG to temperatures as high as 800 151 °C by thermogravimetric analysis (Fig. 4).”;
- In caption of Figure 4, be consistent: “GO(a), and FG (b) by thermogravimetric analysis”;
- Again, fluorinated graphene used in caption of Figure 5 and in text is misleading. It is fluorinated graphene oxide, or reduced fluorinated graphene or equivalent;
- Reformulate: “FG was synthesized with tunable C/F contents and a tunable chemical bonding state with semi-ionic and covalent bonding via a hydrothermal method.”;
- References are not in the right format ([] in text; superscript in the References section);
Author Response
1.In the abstract, line 10, graphene should be used as “reduced graphene” (or “the GO surface”?), in order to avoid further misunderstandings;
=> We apologize this confusion. The content has been revised, and reviewer can confirm the details in the manuscript.
2.FG(F) and FG(HU) needs to be defined or explained in the abstract, right now it is not possible to understand the abstract right away, without reading the main text;
=> The content has been revised, and reviewer can confirm the details in the manuscript.
3.Line 21, “it” is not needed;
=> We apologize this problem. The manuscript was edited by professional English editing service with native speaker. Details can be found in manuscript.
4.I have red with pleasure the introduction part, which is nice, concise, not lengthy and still comprehensible. Nonetheless, some statements could be strengthened by additional references. I believe it could be improved to offer the readers a more perspective view on graphene varieties. Therefore, a reference or two after “properties” in line 22 are needed (maybe Science 2008, 321, 385-388; and/or Solid State Commun. 2008, 146, 351-355 or similar) so that all readers can find those properties and learn about them (otherwise how can the reader compare these properties with respect to properties of F-reduced-graphene in lines 24-26); the paper is intended for all readers not only experienced researchers who already have that kind of information. Furthermore, the expression “Among the numerous graphene derivatives” has to be more explanatory and followed by references, otherwise it is just a very general and meaningless statement with no support from the literature. Maybe: “Among the numerous graphene derivatives, that include GO [Ref: ACS Appl. Mater. Interfaces 2017, 9, 43393-43414 or similar] and RGO [ Mater. Chem. C, 2020, 8, 1198 or similar], fluorinated graphene (FG) is of interest because …..”. Authors should avoid using twice “attracting”, as it is “diluting” the quality of the text. Important: before starting the second sentence “The F atoms ….” in line 23, the authors should add a short, concise sentence explaining how FG, a reduced type of graphene [use Ref: 12 from your text or similar], is different from a “regularly” non-fluorinated/reduced graphene oxide [use Ref: Synthetic Metals, 2020, 269, 116576 or similar].
=> We appreciate reviewer’s comment. We reflected reviewer’s comment and revised manuscript. Details can be found in manuscript.
5.“Along with the synthesis, there are also tried to 33 control the degree of fluoridation” needs reformulation.
=> We apologize this problem. The manuscript was edited by professional English editing service with native speaker. Details can be found in manuscript.
6.H2SO4 should be H2SO4 in line 53; see also for H2O2, KMnO4and MNO2, etc.’
=> We apologize this problem. The experimental section was revised and details can be found in manuscript.
7.“was removed” in line 54;
=> We apologize this problem. The manuscript was edited by professional English editing service with native speaker. Details can be found in manuscript.
8.Reformulate: “After reaction, the white product synthesized in solution, which is FG.”;
=> We apologize this problem. The manuscript was edited by professional English editing service with native speaker. Details can be found in manuscript.
9.Reformulate: “The C 1s spectra also observed to confirm reduction and fluorination of FG (Fig. 3).”;
=> We apologize this problem. The manuscript was revised and details can confirm in manuscript.
10.It looks bad to use in same text Figure, Fig. and FiG. Please fix this through the whole text;
=> We apologize this problem. The manuscript was revised and details can confirm in manuscript.
11.There is no text describing/referring to results in figure 2d;
=> We apologize this problem. The manuscript was revised and details can confirm in manuscript.
12.Sentence “FG(F) and FG(P) exhibit higher C=O contents than FG(HU).” in line 136 seems to contradict with Figure 3. Double check etxt, FG(P) seems to have a smaller C=O peak! Double check the caption of Figure 3, (c) and (d) are missing, (d) is not described at all. Same for Figure 2.
=> We apologize this problem. The manuscript was revised and details can confirm in manuscript.
13.“lower than” in line 137;
=> We apologize this problem. The manuscript was revised and details can confirm in manuscript.
14.Again, there is no text referring/describing Figure 3d;
=> We apologize this problem. The manuscript was revised and details can confirm in manuscript.
15.Reformulate: “The thermal stability of FG to temperatures as high as 800 151 °C by thermogravimetric analysis (Fig. 4).”;
=> We apologize this problem. The manuscript was revised and details can confirm in manuscript.
16.In caption of Figure 4, be consistent: “GO(a), and FG (b) by thermogravimetric analysis”;
=> We apologize this problem. The manuscript was revised and details can confirm in manuscript.
17.Again, fluorinated graphene used in caption of Figure 5 and in text is misleading. It is fluorinated graphene oxide, or reduced fluorinated graphene or equivalent;
=> We apologize this problem. The manuscript was revised and details can confirm in manuscript.
18.Reformulate: “FG was synthesized with tunable C/F contents and a tunable chemical bonding state with semi-ionic and covalent bonding via a hydrothermal method.”;
=> We apologize this problem. The manuscript was revised and details can confirm in manuscript.
19.References are not in the right format ([] in text; superscript in the References section);
=> We apologize this problem. The manuscript was revised and details can confirm in manuscript.

Reviewer 3 Report
In this work by Lee et al., the authors studied the fluorination of graphene oxide. The results fit the scope of Nanomaterials, but the study contains significant shortcomings which should be addressed before the paper can be re-evaluated by the journal. Please find the comments below:
1) "FG(F) and FG(HU) exhibited less extensive fluorination than FG(P) despite their same or greater oxygen content compared with that of FG(P)" This sentence in the introduction is unclear as, at this point, the readers do not know what these abbreviations mean. A similar mistake is present at the end of the abstract.
2) "Recent research has demonstrated that FG has unique chemical, physical, mechanical and electrical properties compared to graphene" - specify what properties and provide some quantification. At present, this sentence carries no value, especially that it is not supported with any references.
3) The state of the art is poorly described. There are just 12 references in the Introduction section.
4) "In this work" part of the introduction is too short as it does not summarize well the article. Besides that, the novelty factor is not clearly defined. Please specify directly what is new in this paper that was not done by others before.
5) The experimental section is not written with sufficient scientific rigor. Consequently, it is not possible to validate these findings and build on them. To avoid a limited impact of this article, all the necessary parameters should be supplemented. Examples of shortcomings are given below. For the sake of brevity, I will not list all of them. I invite the authors to carefully screen this section.
- "Three types of GO with different oxygen binding states were used in the present study" - what does it mean?
- "5g of graphite and 2.5 g of NaNO3 were stirred with H2SO4 in an ice bath. 15g of KMnO4 was slowly added in solution. " - what was the amount of H2SO4?
- centrifugation parameters not given
- "The as-prepared FG was filtered with micro pore filter " - what was the porosity exactly?
- "washed with DI water several times, and then freeze-dried" - what were the freeze-drying conditions?
- characterization parameters not reported
- etc.
6) The level of English should be improved considerably.
7) TEM micrographs are barely visible. Besides, the starting material should be characterized as well.
8) Why is there no CF2 component in Fig. 2c?
9) How is that the positions of the peak maxima in Fig. 2 are not consistent? This is not an appropriate way to carry out XPS analysis.
10) Similar questions can be asked about Fig. 3. Please comment on this issue.
11) Thermograms of starting GO materials are not described.
12) Taman spectra of non-fluorinated GO materials are not enclosed.
13) There is no thorough analysis of the reported results. Most of them are simply listed without any convincing hypothesis from the authors or the literature.
Author Response
In this work by Lee et al., the authors studied the fluorination of graphene oxide. The results fit the scope of Nanomaterials, but the study contains significant shortcomings which should be addressed before the paper can be re-evaluated by the journal. Please find the comments below:
1) "FG(F) and FG(HU) exhibited less extensive fluorination than FG(P) despite their same or greater oxygen content compared with that of FG(P)" This sentence in the introduction is unclear as, at this point, the readers do not know what these abbreviations mean. A similar mistake is present at the end of the abstract.
=> We apologize for this problem. We reflected reviewer’s comment and the details can confirm in the manuscript.
2) "Recent research has demonstrated that FG has unique chemical, physical, mechanical and electrical properties compared to graphene" - specify what properties and provide some quantification. At present, this sentence carries no value, especially that it is not supported with any references.
=> We apologize for this problem. We have revised the manuscript and the details can confirm in the manuscript.
3) The state of the art is poorly described. There are just 12 references in the Introduction section.
=> We apologize for this problem. We have revised the manuscript and the details can confirm in the manuscript.
4) "In this work" part of the introduction is too short as it does not summarize well the article. Besides that, the novelty factor is not clearly defined. Please specify directly what is new in this paper that was not done by others before.
=> We have revised the manuscript. Details can be confirmed in the manuscript.
5) The experimental section is not written with sufficient scientific rigor. Consequently, it is not possible to validate these findings and build on them. To avoid a limited impact of this article, all the necessary parameters should be supplemented. Examples of shortcomings are given below. For the sake of brevity, I will not list all of them. I invite the authors to carefully screen this section.
- "Three types of GO with different oxygen binding states were used in the present study" - what does it mean?
- "5g of graphite and 2.5 g of NaNO3 were stirred with H2SO4 in an ice bath. 15g of KMnO4 was slowly added in solution. " - what was the amount of H2SO4?
- centrifugation parameters not given
- "The as-prepared FG was filtered with micro pore filter " - what was the porosity exactly?
- "washed with DI water several times, and then freeze-dried" - what were the freeze-drying conditions?
- characterization parameters not reported
- etc.
=> We apologize for any confusion on this problem. The properties of GO can confirm in the manuscript and the denoted type also written in manuscript. The manuscript has been revised and details can be confirmed in the manuscript.
6) The level of English should be improved considerably.
=>We apologize for this problem. The manuscript was edited by professional English editing service with native speaker. Details can be confirmed in manuscript.
7) TEM micrographs are barely visible. Besides, the starting material should be characterized as well.
=> We apologize for this problem. The GO characterized FT-IR, XPS, Raman, and TG. TEM image try was improved.
8) Why is there no CF2 component in Fig. 2c?
=> GO(HU) showed low C=O and C–O–C bonding on the graphene plane compared to GO(F) and GO(P), as observed in the FT-IR spectra. Therefore, the strong covalent bonding of C-F2 was not found.
9) How is that the positions of the peak maxima in Fig. 2 are not consistent? This is not an appropriate way to carry out XPS analysis.
=>We confirmed the XPS data again carefully. There was an error in the merge process with other data. The manuscript has been revised, and details can be found in the manuscript.
10) Similar questions can be asked about Fig. 3. Please comment on this issue.
=>We confirmed the XPS data again carefully. There was an error in the merge process with other data. The manuscript has been revised, and details can be found in the manuscript.
11) Thermograms of starting GO materials are not described.
=> We apologize for this problem. The manuscript has been revised and details can be confirmed in the manuscript.
12) Raman spectra of non-fluorinated GO materials are not enclosed.
=> The GO characterized Raman. The manuscript has been revised and details can be confirmed in the manuscript.
13) There is no thorough analysis of the reported results. Most of them are simply listed without any convincing hypothesis from the authors or the literature.
=> We appreciate reviewer’s comment. . The manuscript was revised to reflect the reviewer's opinion.

Reviewer 4 Report
The manuscript describes a study about the Fluorinate Graphene synthesized with tunable C/F contents and a tunable chemical bonding state with semi-ionic and covalent bonding via a hydrothermal method. The different O content and chemical state on the GO surface results in differences of fluorination and the extent of reduction. The fluorination reaction also helps to improve the reduction degree of the GO. The high concentration of C=O in GO promotes the formation of covalent bonds of C–F, C–F2, and C–F3 in FG. Oxygen in other states such as –COOH results in semi-ionic bonding. The reaction mechanism of the fluorination is clarified by tracking the oxygen bonding state in the initial GO. The doped F atoms tune the electronic and thermal properties and surface chemistry of graphene.
The manuscript shows a remarkably interesting experimental results, clearly presented and organized. The introduction such as the discussion is well supported by interesting references. However, the authors must improve the experimental information about the three type of GO used as precursors. To better analyze the effect of oxidation grade of GO on the synthesis FG they must show a very extensive characterization of the GO precursors (Type I (GO(F)) and Type II ((GO(P)), and Type III ((GO(HU)) and they must clarify the different state of oxidation by FT-IR and improve with this information the discussion of experimental results. In addition, English language and style are fine/minor spell check required, for example a lot of Chemical formula need use of subscript.
For this the manuscript can be accepted after major revision.
Author Response
The manuscript shows a remarkably interesting experimental results, clearly presented and organized. The introduction such as the discussion is well supported by interesting references. However, the authors must improve the experimental information about the three type of GO used as precursors. To better analyze the effect of oxidation grade of GO on the synthesis FG they must show a very extensive characterization of the GO precursors (Type I (GO(F)) and Type II ((GO(P)), and Type III ((GO(HU)) and they must clarify the different state of oxidation by FT-IR and improve with this information the discussion of experimental results. In addition, English language and style are fine/minor spell check required, for example a lot of Chemical formula need use of subscript.
For this the manuscript can be accepted after major revision.
=> We appreciate reviewer’s comment. We investigated the GO surface properties by FT-IR and revised the experimental section. The manuscript was edited by professional English editing service with native speaker. Details can be found in manuscript.

Round 2
Reviewer 1 Report
The authors adequately addressed the comments. A big improvement is the consideration of the C=O IR peak analysis to supplement evidence for the conclusions. Language was improved owing to official language check, which was very necessary. I can suggest publication.
Author Response
The manuscript was carefully reviewed by the MDPI English Proofreading Service. We have attached the confirmation file.

Reviewer 3 Report
The manuscript was improved, but several comments were disregarded. For instance:
Remark #5:
- "Three types of GO with different oxygen binding states were used in the present study" - what does it mean? - NOT answered
- centrifugation parameters not given - NOT answered. Please give the parameters in rcf values.
- "The as-prepared FG was filtered with micro pore filter " - what was the porosity exactly? - NOT answered
- characterization parameters not reported - XPS parameters should be described more precisely. Please include information on how the background was selected and subtracted. Also, include how the components were deconvoluted.
Remark #8: Yet, the CF2 component is definitely present as gauged by XPS since an appropriate peak can be produced in this area. Please consider deconvoluting Fig. 4c to three components to make the analysis consistent, especially since that the presence of the CF2 feature is obvious in this area. Its intensity is even higher than in Fig. 4a, where it was assigned. If corrected, Fig. 5 should be modified accordingly.
Remark #12: Please describe why such a tremendous improvement to thermal stability in Fig. 6 was observed. In addition, "The significant weight loss under 200 °C of GOs (Fig. 7a), which can be attributed to the elimination of the water molecules and unstable oxygen groups." Samples should be kept in a desiccator prior to analysis to eliminate this effect.
Captions should not be separated from the corresponding images e.g. Fig. 1.
Besides, there is lots of redundant empty space, which should be removed.
Author Response
The manuscript was improved, but several comments were disregarded. For instance:
Remark #5:
- "Three types of GO with different oxygen binding states were used in the present study" - what does it mean? - NOT answered
=> The three types of GO denoted GO(F), GO(P) and GO(HU) in manuscript. The different oxygen binding states means different oxygen functional groups. We changed expression in manuscript.
- centrifugation parameters not given - NOT answered. Please give the parameters in rcf values.
=>The condition of centrifugation is 10000 rpm during 30min at once and the process repeated with 5 times
=> GOs was dispersed in DI water by ultrasonication (pulse 10s on and 5s off, amplitude 20%, 20kHz, 500W, VC500, Sonics & Materials, USA) for 30 min under 10 °C.
- "The as-prepared FG was filtered with micro pore filter " - what was the porosity exactly? - NOT answered
=> The micro pore filter used 0.45 ㎛.
- characterization parameters not reported - XPS parameters should be described more precisely. Please include information on how the background was selected and subtracted. Also, include how the components were deconvoluted.
=> XPS spectra characterized by a monochromatic Al Kα source. The peak was analyzed using peak analyzer via origin software. Curve fitting of the spectra was performed using a Gaussian peak fitting after conducting a Shirlely background correction.
Remark #8: Yet, the CF2 component is definitely present as gauged by XPS since an appropriate peak can be produced in this area. Please consider deconvoluting Fig. 4c to three components to make the analysis consistent, especially since that the presence of the CF2 feature is obvious in this area. Its intensity is even higher than in Fig. 4a, where it was assigned. If corrected, Fig. 5 should be modified accordingly.
=> As advised by the reviewer, I tried to deconvolute the peak for CF2 in FG (HU) (Fig. 4c), but an error continuously occurred. One of the trials showed possible peak formation with very low intensity, but no more peaks appeared at the C1s peak. Therefore, we think that the peak we have presented is valid.
Remark #12: Please describe why such a tremendous improvement to thermal stability in Fig. 6 was observed. In addition, "The significant weight loss under 200 °C of GOs (Fig. 7a), which can be attributed to the elimination of the water molecules and unstable oxygen groups." Samples should be kept in a desiccator prior to analysis to eliminate this effect.
=> FG began to decompose in the temperature range of 300–400 °C. In decomposition at higher temperatures (400–600 °C), FGs are converted into low-molecular-weight volatile compounds (CxFy). FG shows good thermal stability as a result of strong C–F bonding energy. FG(P) shows higher weight loss compared to others. It suggests that the structure is unstable due to high fluorination and is easily changed to a volatile compound (CxFy). On the other hand, FG (F) and FG (HU) have a relatively stable structure compared to FG (P) and are considered to maintain the advantages of fluorination while maintaining thermal stability. The fluorination degree was determined by the oxygenation state, which strongly influenced the chemical composition and structure of the final FG. Consequently, it was confirmed that the chemical properties of graphene affect the fluorination and, conclusively, the thermal properties.
Captions should not be separated from the corresponding images e.g. Fig. 1.
=> We adjust figure 1 size and the caption is not separated.
Besides, there is lots of redundant empty space, which should be removed.
=> We revised manuscript.

Reviewer 4 Report
The manuscript can be published in present form.
Author Response
We appreciate your comment.